# Effects of Service Quality Characteristics of Neighborhood Sports Facilities on User Satisfaction and Reuse Intention of the Elderly during the COVID-19 Pandemic

**DOI:** 10.3390/ijerph192114606

**Published:** 2022-11-07

**Authors:** Kyoung Hwan Cho, Jeong-Beom Park, Yang Hun Jung

**Affiliations:** 1Department of Special Physical Education, Daelim University College, Anyang-si 13916, Korea; 2Department of Special Physical Education, Hanshin University, Osan-si 18101, Korea

**Keywords:** COVID-19, the elderly, neighborhood sports facilities, service quality characteristics, user satisfaction, reuse intention

## Abstract

Purpose: this study aimed to examine the user satisfaction and reuse intention of the elderly for neighborhood sports facilities in South Korea during the COVID-19 pandemic. Methods: this study surveyed 386 Korean elderly individuals aged ≥ 65 years, who were users of neighborhood sports facilities, from 1 May to 31 August 2022. A total of 386 questionnaires were used for data analysis, which was carried out using SPSS 23.0 statistical software. Descriptive statistics of the mean, standard deviation, and frequency distribution were used at the descriptive level. Moreover, one-way analysis of variance (ANOVA), Scheffe’s post hoc pair-wise comparison analysis, Pearson’s correlation coefficient, and multiple regression analysis were used at the inferential level. The significance level of these tests was considered for less than 0.05. Results: the mechanistic and humanistic service factors of neighborhood sports facilities affected user satisfaction and reuse intention. Furthermore, user satisfaction of the elderly during the COVID-19 pandemic had a positive effect on reuse intention. Conclusion: this study confirmed that the service quality characteristics of neighborhood sports facilities during the COVID-19 pandemic had a positive effect on user satisfaction and intention to continue to exercise among the elderly.

## 1. Introduction

With rapidly rising human life expectancy [1], South Korea (hereafter ‘Korea’) is witnessing the most rapid population aging among Asian countries [2]. The elderly population in Korea continues to grow with extending life expectancy [3]. In addition to rapidly increasing longevity, Korea is also experiencing demographic aging as a result of declining fertility rates, and is thus globally regarded as the most rapidly aging population [4]. Older adults experience varied changes in different aspects of their lives [5]. Physiological decline that accompanies aging involves physical changes that occur as a result of a decline in normal functioning of the body, such as poor mobility, hearing and vision; inability to properly eat and digest food; poor memory; inability to control certain physiological functions; and various chronic conditions [6]. Exercise has been shown to have a positive impact on a variety of physiological and psychological functions among the elderly [7]. Regular physical exercise can mitigate age-related functional decline [8]. Physical activity is important for the elderly, especially for maintaining their independence, mental health, and well-being [9]. However, the worldwide spread of COVID-19 has resulted in a global health crisis [10]. Due to the COVID-19 pandemic, many countries temporarily closed fitness centers and gymnasiums, before vaccines were available to the public [11]. Therefore, the COVID-19 pandemic became a major threat to public health [12]. Moreover, the elderly, and those with pre-existing medical conditions, are at risk of death from COVID-19 [13]. Even during the COVID-19 pandemic, the elderly require regular exercise to prevent muscle weakness; for that, they use neighborhood sports facilities, which are outdoor sports facilities that are located in parks or within the village, as well as private or public gymnasiums [14]. It is necessary to examine the perception of service quality, user satisfaction, and revisit intention of the elderly, who frequently use neighborhood sports facilities during the COVID-19 pandemic. Therefore, this study aimed to provide basic data for the expansion of facilities that contribute to the health of the elderly, by examining the effect of service quality of neighborhood sports facilities on the satisfaction and reuse intention of elderly users.

## 2. Materials and Methods

### 2.1. Research Design

This study used the SERVQUAL model to verify the hypotheses, which was developed by Parasuraman, Zeithaml, and Berry [15]. SERVQUAL is an instrument to measure quality that stems from this model, and works via differences in scores (gaps) that are obtained from questionnaires [16]. Service quality that is related to neighborhood sports facilities refers to mechanistic (objective) quality factors, and humanistic (subjective) quality factors that are provided to users at the sports facility [17]. Thus, service quality of neighborhood sports facilities in this study consisted of both mechanistic and humanistic aspects. In addition, user satisfaction can be defined as a measurement that determines how happy users are with the services that are provided by the sports facilities [18], while reuse intention is defined as the users being willing to continue to use the service from the same sports facilities [19]. Figure 1 shows a research model that assumes the relationship between the service factors of neighborhood sports facilities and user satisfaction or reuse intention. This study established the following hypotheses:

**H1:** 
*The service quality of neighborhood sports facilities shall have a positive effect on user satisfaction.*


**H1-1:** 
*Mechanistic service factors shall have a positive effect on user satisfaction.*


**H1-2:** 
*Humanistic service factors shall have a positive effect on user satisfaction.*


**H2:** 
*The service quality of neighborhood sports facilities shall have a positive effect on reuse intention.*


**H2-1:** 
*Mechanistic service factors shall have a positive effect on reuse intention.*


**H2-2:** 
*Humanistic service factors shall have a positive effect on reuse intention.*


**H3:** 
*User satisfaction shall have a positive effect on reuse intention.*


**Figure 1 ijerph-19-14606-f001:**
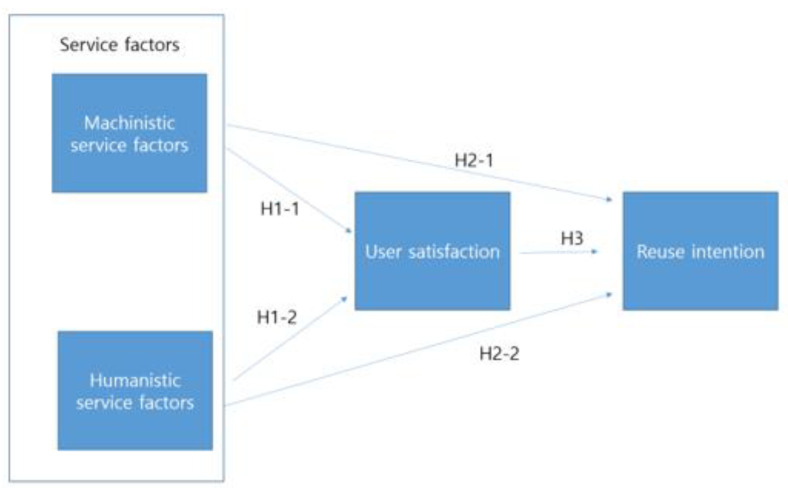
Research model.

### 2.2. Research Subjects

After receiving an explanation regarding the purpose of this study, elderly subjects living in Seoul were selected, who consented to participate, and who used neighborhood sports facilities. The minimum sample size was obtained using the G*power 3.1.9.2 program [20] based on previous studies on user satisfaction and reuse intention for community education facilities [21]. With a significance level of 0.05, power of 0.80, and effect size of 0.30, the minimum sample size was confirmed to be 352 persons. Therefore, a total of 388 participants were recruited, considering a dropout probability of 10%. Among them, 2 responses were deemed insincere (e.g., two or more answers missing) and were consequently excluded, resulting in 386 valid responses for analysis.

### 2.3. Research Instruments

The 22-item survey used in this study consisted of 6 questions on general characteristics, 8 on the selection attribute of the neighborhood sports facilities, 4 on user satisfaction, and 4 on reuse intention. Responses were provided using a Likert 5-point scale. Referring to a previous study [22], this research divided the eight items of service quality into four items, each relating to mechanistic and humanistic qualities. Mechanistic quality factors included objective aspects or characteristics of things or events, while humanistic quality factors included participants’ subjective reactions to objects. Figure 2 shows a neighborhood sports facility.

After referring to a previous study [23], user satisfaction, a dependent variable, was defined as the overall satisfaction formed after using a local sports facility, and four questions were used to assess this characteristic. Moreover, after referring to a previous study [24], reuse intention, a dependent variable, was defined as the intention to continuously use neighborhood sports facilities, and four questions were presented to assess this attribute in the questionnaire.

### 2.4. Reliability and Validity

The main objective of the questionnaire used in this study was to obtain relevant information in the most reliable and valid manner. Reliability and validity are concepts used to evaluate the quality of research [25]. Cronbach’s α was estimated in this study, since it was the most widely used objective measure of reliability. Table 1 shows the Cronbach’s alpha values of mechanistic service factors, humanistic service factors, user satisfaction, and reuse intention were 0.901, 0.873, 0.901, and 0.890, respectively.As a result, the research instruments were qualified for reliability, or for being reliable [26].

In this study, six experts with a doctorate in physical education or statistics were recruited, in order to evaluate the validity of the questionnaire items. Content validity index (CVI) is the most widely used index in quantitative evaluation [27]. The scale-level content validity index (S-CVI) was calculated to demonstrate content validity. Experts were asked to rate each item in the questionnaire on a 4-point Likert scale: 1 (highly irrelevant), 2 (not relevant), 3 (relevant), and 4 (highly relevant) [28].

Scores 1 and 2 were then classified as irrelevant (value of 0), while scores 3 and 4 were classified as relevant (value of 1). S-CVI/Ave (average variance extracted) was calculated to determine each item’s mean item level-CVI(I-CVI). The measurement of content validity depicts the degree to which an item in an assessment instrument is relevant to a representative for a particular purpose. The higher a content validity test score, the more accurate it is in measuring the target construct. I-CVI > 0.79 means that the item can be relevant, and does not need further revision; S-CVI/Ave ≥ 0.9 indicates that the items have excellent content validity [29]. Table 2 shows that the S-CVI/Ave of the research instruments was 0.92. However, the answers of Q1 and Q5, whose I-CVIs were below 0.79, were excluded from the analysis, in order to maintain the validity of this study.

### 2.5. Data Collection and Analysis Methods

This study conducted a convenience sampling of Korean men and women aged 65 years and over who had experience at neighborhood sports facilities during the COVID-19 pandemic. In this study, random sampling was performed on the elderly living in Seoul. Written consent was obtained from the respondents, and data were collected using a structured questionnaire. A total of 386 questionnaires were used for data analysis, which was carried out using SPSS v22.0 software (IBM Corp., Somers, NY, USA). Descriptive statistics of the mean, standard deviation, and frequency distribution were used at the descriptive level. Moreover, one-way analysis of variance (ANOVA), Scheffe’s post hoc pair-wise comparison analysis, Pearson’s correlation coefficient, and multiple regression analysis were used at the inferential level. The significance level of these tests was considered for less than 0.05.

## 3. Results and Discussion

### 3.1. Descriptive Statistics

Table 3 shows the demographic characteristics of the respondents: 284 (73.6%) were male, 361 (93.5%) were married, 361 (93.5%) were between 65 and 74, 189(48.9%) had graduated from college, and 365 (94.6%) assumed that they were healthy. Moreover, 194 (50.3%) used neighborhood sports facilities once a week or less.

Table 4 presents the differences in mechanistic service factors, according to demographic characteristics. The respondents had different mechanistic service factors, according to the marital status. Males (3.56) had significantly higher user satisfaction than females (3.01) (t = 3.809, *p* < 0.001). The mechanistic service factors of “graduated from college” group (3.60) and “graduated from graduate school or higher” group (3.94) were higher than those of “graduate from high school or less” group (3.25), which was statistically significant (F = 22.260, *p* < 0.001). Scheffe’s post hoc test showed that their mechanistic service factors were statistically different, in the order of “graduated from high school or less,” “graduated from college,” and “graduated from graduate school or higher” groups. The “6 or more times per week” group (4.69) had significantly higher mechanistic service factors than those of other groups (F = 18.948, *p* < 0.001), which was verified by Scheffe’s post hoc test.

Table 5 presents the differences in humanistic service factors, according to demographic characteristics. The respondents had different humanistic service factors, according to their gender. Males (2.73) had significantly higher humanistic service factors than females (2.35) (t = 4.275, *p* < 0.001). Furthermore, their levels of humanistic service factors were different, according to the health status of the respondents. Humanistic service factors in the “healthy” group (2.69) were significantly higher than those in the “very healthy” group (1.59) (t = 5.892, *p* < 0.001). According to the age of the respondents, they had statistically different humanistic service factors (F = 34.616, *p* < 0.001). Scheffe’s post hoc test showed that humanistic service factors were significantly higher, in the order of the 65–74 age group (2.71), the 75–84 age group (1.97), and the 85 or more age group (1.14).

The “graduated from college” group had higher humanistic service factors than the groups with other educational backgrounds (F = 5.799, *p* < 0.01), and Scheffe’s post hoc test showed that their humanistic service factors (2.77) were higher than those of the “graduated from high school or less” group (2.44). The “4–5 times a week” group (2.97) had higher humanistic service factors than those of other groups (F = 15.198, *p* < 0.001). However, Scheffe’s post hoc test showed that the result was significant when the “4–5 times a week” group was compared with the “once a week or less” group.

Table 6 presents the differences in user satisfaction according to demographic characteristics. The respondents had different levels of user satisfaction according to their gender. That is, males (3.37) had significantly higher user satisfaction than females (3.16) (t = 2.049, *p* < 0.05). Furthermore, their levels of user satisfaction were different according to the health status of the respondents. The healthy group’s user satisfaction (3.36) was significantly higher than that of the very healthy group (2.24) (t = 6.033, *p* < 0.001). The 65–74 age group (3.38) and the 75–84 age group (3.41) had higher levels of user satisfaction than the 85 or more age group (1.43) (F = 66.078, *p* < 0.001), which was verified by Scheffe’s post hoc test. The “graduated from college” group (3.39) and the “graduated from graduate school or higher” group (3.39) had higher levels of user satisfaction than the “graduated from high school” group (3.18) (F = 3.712, *p <* 0.05), which was verified by Scheffe’s post hoc test. The “2–3 times a week” group (3.71) and the “4–5 times a week” group (3.61) had higher levels of user satisfaction than the “once a week or less” group (2.98) (F = 15.198, *p <* 0.001), which was verified by Scheffe’s post hoc test. However, the marital status of the respondents did not have a significant effect on user satisfaction.

Table 7 presents the differences in reuse intention according to demographic characteristics. The respondents had different levels of user satisfaction according to their gender. That is, males (3.35) had significantly higher user satisfaction than females (3.06) (t = 3.060, *p* < 0.001). Moreover, their levels of reuse intention were different according to the health status of the respondents. The healthy group’s user satisfaction (3.93) was significantly higher than that of the very healthy group (2.33) (t = 4.661, *p* < 0.001). The 65–74 age group (3.34) and the 75–84 age group (3.45) had higher levels of user satisfaction than the 85 or more age group (1.43) (F = 56.267, *p <* 0.001), which was verified by Scheffe’s post hoc test. The “2–3 times a week” group (3.59) and the “4–5 times a week” group (3.59) had higher levels of user satisfaction than the “once a week or less” group (2.99) (F = 23.024, *p* < 0.001), which was verified by Scheffe’s post hoc test. However, the educational background or marital status of the respondents did not have a significant effect on reuse intention.

### 3.2. Correlation Analysis

Table 8 shows the Pearson correlation analysis results between variables. There was a positive correlation between mechanistic service factors and user satisfaction (r = 0.488, *p* < 0.01). Furthermore, there was a positive correlation between humanistic service factors and user satisfaction (r = 0.625, *p* < 0.01). In particular, humanistic service factors correlated more with user satisfaction than mechanistic service factors. There was a positive correlation between mechanistic service factors and reuse intention (r = 0.365, *p* < 0.01). Moreover, there was a positive correlation between humanistic service factors and reuse intention (r = 0.507, *p* < 0.01). Furthermore, humanistic service factors correlated more with reuse intention than mechanistic service factors. Finally, there was a positive correlation between user satisfaction and reuse intention (r = 0.868, *p* < 0.01).

### 3.3. Validation of the Research Hypotheses

#### 3.3.1. Validation of H1 Hypothesis

Multiple linear regression analysis was performed, in order to examine whether mechanistic service factors or humanistic service factors significantly predicted user satisfaction.

Table 9 shows that the predictors of the regression model explained 50.6% of the variance, and a collective significant effect was found (F = 196.380, *p* < 0.001, R^2^ = 0.506). In particular, it was confirmed that humanistic service factors (β = 0.535, *p* < 0.001) had a higher influence on user satisfaction than mechanistic service factors (β = 0.352, *p* < 0.001). As a result, service factors of neighborhood sports facilities affected user satisfaction (H1 is accepted). Mechanistic service factors and humanistic service factors also affected user satisfaction (H1-1 and H1-2 are also accepted, respectively).

#### 3.3.2. Validation of H2 Hypothesis

Multiple linear regression analysis was also performed, in order to examine whether mechanistic service factors or humanistic service factors significantly predicted reuse intention. Table 10 presents that the predictors of the regression model explained 31.7% of the variance, and a collective significant effect was found (F = 88.869, *p* < 0.001, R^2^ = 0.317). In particular, humanistic service factors (β = 0.443, *p* < 0.001) had a higher influence on reuse intention than mechanistic service factors (β = 0.253, *p* < 0.001). As a result, service factors of neighborhood sports facilities affected reuse intention (H2 is accepted). Mechanistic service factors and humanistic service factors also affected reuse intention (H2-1 and H2-2 are also accepted, respectively).

#### 3.3.3. Validation of H3 Hypothesis

Linear regression analysis was performed, in order to examine whether user satisfaction significantly predicted reuse intention. The regression model indicated that the predictors explained 75.3% of the variance, and a collective significant effect was found (F = 1169.670, *p* < 0.001, R^2^ = 0.753). In other words, user satisfaction had a significant effect on reuse intention (β = 0.868, t = 34.200, *p* < 0.001). Therefore, H3 was accepted (Table 11).

## 4. Discussion

Neighborhood sports facilities in large cities were located outdoors, providing opportunities for exercise to many elderly people in extraordinary situations, where the use of indoor facilities was limited during the COVID-19 pandemic. Thus, this study aimed to analyze the effects of service quality characteristics of neighborhood sports facilities on user satisfaction and reuse intention of the elderly during the COVID-19 pandemic in Korea. This study surveyed 386 individuals aged 65 or older living in Korea. This study measured the validity and reliability of the research instruments by referring to previous studies [30,31,32]. Two items with low validity were deleted, and the collected data were analyzed using a statistical program. This study classified the service quality of sports facilities into mechanistic and humanistic factors [17]. In this study, the mechanistic service factors of neighborhood sports facilities included transportation, proximity, and accessibility, while humanistic service factors included sports programs, information, and medical support.

This study approached the service factors of neighborhood sports facilities by classifying them into mechanistic and humanistic service factors. Mechanistic service factors were measured to be high in variables such as male (3.57), married (3.56), healthy (3.53), 65–74 age group (3.53), “graduated from graduate school or higher” (3.94), and “2–3 times per week” (3.61). Humanistic service factors were measured to be high in variables such as male (2.73), healthy (2.71), “graduated from college” (2.77), and “4–5 times per week” (2.91). Levels of user satisfaction were measured to be high in variables such as male (3.37), healthy (3.36), 65–74 age group (3.38) or 75–84 age group (3.41), “graduated from college or graduate school” (3.39), and “2–3 times per week” (3.71). Levels of reuse intention were measured to be high in variables such as male (3.35), healthy (2.69), 65–74 age group (2.71), and “2–3 or 4–5 times per week” (3.59). Thus, it was confirmed that the elderly’s perception of service factors of neighborhood sports facilities differed according to gender, age, education level, health status, and frequency of visits to facilities.

The Pearson correlation analysis showed positive correlations among the service factors, user satisfaction, and reuse intention. Thus, this study tested three hypotheses on the relationships between service factors, user satisfaction, and reuse intention of neighborhoods. The results showed that service factors of neighborhood sports facilities affected user satisfaction. Moreover, mechanistic service factors and humanistic service factors affected user satisfaction. In addition, this study revealed that service factors of neighborhood sports facilities affected reuse intention. Mechanistic service factors and humanistic service factors also affected reuse intention. Finally, user satisfaction significantly affected reuse intention (β = 0.868, t = 34.200, *p* < 0.001). Based on the above results, this study discusses the points that follow.

Firstly, this study revealed that the service factors of neighborhood sports facilities used by the elderly during the COVID-19 pandemic had a positive effect on user satisfaction. Previous research [33,34] also reported a significant positive relationship between service quality and user satisfaction, thus supporting the results of the present study. This study demonstrated that humanistic service factors (β = 0.443, *p* < 0.001) had a higher influence on reuse intention than mechanistic service factors (β = 0.253, *p* < 0.001). In addition, humanistic service factors (β = 0.443, *p* < 0.001) had a higher influence on reuse intention than mechanistic service factors (β = 0.253, *p* < 0.001). This suggests that there is a crucial role for exercise professionals in the operation of neighborhood sports facilities [35]. Therefore, among the service qualities of neighborhood sports facilities, humanistic service factors should be improved more than mechanistic service factors, and the quality of information and medical services—two factors with below average scores—should be improved.

Furthermore, the mechanistic service factors of neighborhood sports facilities are still insufficient in comparison to the service factors of general and private sports facilities. As a result, careful consideration, attention, and policy planning from the local and state governments are required. That is, the administrative body that is responsible for the operation and management of neighborhood sports facilities should propose plans for further improvements of service factors, in order to promote the elderly’s health and quality of life. However, most neighborhood sports facilities are operated by public institutions, which are often reported to have a lower ability to maintain customer satisfaction than private institutions [36]. As a result, it is necessary for public institutions that operate neighborhood sports facilities to check user satisfaction, and to subsequently make efforts to increase it.

Secondly, the finding that the service factors of neighborhood sports facilities directly affects reuse intention is consistent with previous studies [19,37,38]. Therefore, it is crucial to improve the service quality, in order to promote the physical activity of the elderly in neighborhood sports facilities, and to increase their reuse or revisit [38]. In other words, it can be suggested that the specialization, diversification, and upgrading of neighborhood sports facilities can increase the number of user revisits. This study confirmed that the elderly frequently used outdoor sports facilities near their homes during the COVID-19 pandemic. However, this phenomenon may be specific to the context of COVID-19. That is, if indoor sports facilities are normalized in the post-corona era, elderly users may move to private sports facilities that have better service quality [19]. Therefore, user satisfaction and reuse intention among the elderly can be maintained and enhanced, only when efforts to improve the service factors of neighborhood sports facilities continue.

Thirdly, user satisfaction in this study showed a significant effect on reuse intention, supporting the results of previous studies [19,39] which reported that users with high satisfaction with sports facilities have a high reuse intention, and moreover, also recommend the facilities to others. Thus, this study shows that improving service factors of neighborhood sports facilities can increase reuse intention, as mediated by user satisfaction. As a result, user satisfaction with neighborhood sports facilities can be seen as a factor that has a great influence on reuse intention after facility use; therefore, public institutions in charge of operating neighborhood sports facilities must seek to enhance the administration and management of the facilities, and improve service quality. Furthermore, efforts to improve service quality should be continued so that the elderly can maintain their health by using neighborhood sports facilities, even in the post-corona era.

Fourthly, this study has a limitation in that it was conducted on the elderly living in Seoul during the COVID-19 pandemic. The elderly who live in Seoul have many opportunities to use fitness centers and indoor gymnasiums. Therefore, when the COVID-19 pandemic is over, the elderly in Seoul can exercise indoors rather than using neighborhood sports facilities. Meanwhile, the elderly, who live in small towns or rural areas where there are not many indoor sports facilities, may have a higher possibility of using neighborhood sports facilities. In a follow-up study, it will be helpful to generalize the research results by separately approaching the elderly in rural versus urban areas. Furthermore, examining the differences in the perceptions of the elderly about neighborhood living facilities, according to gender, could provide more meaningful research results.

## 5. Conclusions

In conclusion, this study confirmed that the mechanistic and humanistic service factors of neighborhood sports facilities during the COVID-19 pandemic had a positive effect on the elderly’s user satisfaction and reuse intention. If mechanistic and humanistic service factors are not improved, the elderly may not be satisfied with neighborhood sports facilities, or may not reuse them in the post-corona era. Therefore, this study has some implications, in that it is necessary to maintain the service quality factors of the neighborhood sports facilities to increase user satisfaction and reuse intention. However, this study has a limitation in that it did not analyze potential mediating or moderating variables in relation to service factors of the neighborhood sports facilities, user satisfaction, and reuse intention. Therefore, the need to identify potential mediating or moderating effects among the variables, using structural equation modeling in follow-up studies, is suggested.

## Figures and Tables

**Figure 2 ijerph-19-14606-f002:**
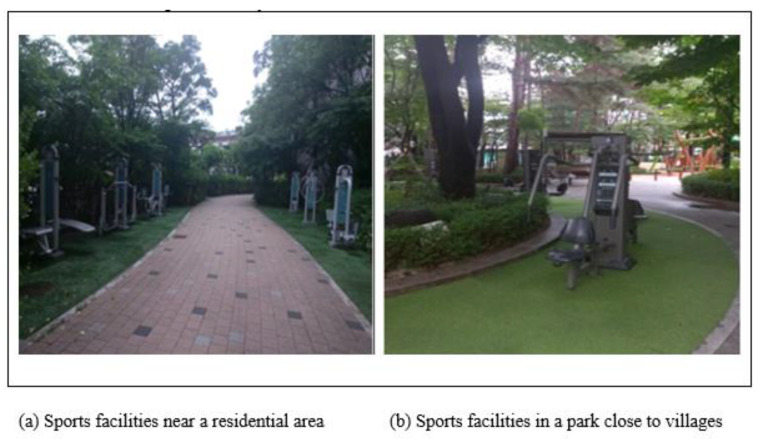
Neighborhood sports facilities. In Korea, neighborhood sports facilities are mainly located near residential areas (**a**) or in parks close to villages (**b**).

**Table 1 ijerph-19-14606-t001:** Results for reliability concerning measurement instruments.

Category		Cronbach α
Service factors	Mechanistic service factors	0.901
	Humanistic service factors	0.873
User satisfaction		0.901
Reuse intention		0.890

**Table 2 ijerph-19-14606-t002:** Ratings of item relevance of six experts.

Factors	Item	Experts in Agreement (*n* = 6)	I-CVI ^a^	Results
Mechanistic service factors	Q1. Is the scenery around the neighborhood sports facilities good?	3	0.50	Rejected
	Q2. Is transportation to and from the neighborhood sports facility convenient?	5	0.83	Accepted
	Q3. Is the neighborhood sports facility close to the promenade?	6	1.00	Accepted
	Q4. Is it easy to access the neighborhood sports facility?	6	1.00	Accepted
Humanistic service factors	Q5. Do you feel safe about the facility?	3	0.50	Rejected
	Q6. Is the facility’s exercise program appropriate?	5	0.83	Accepted
	Q7. Is information on the facility provided?	6	1.00	Accepted
	Q8. Is appropriate medical treatment provided in case of injury at the facility?	6	1.00	Accepted
User satisfaction	Q9. I am overall satisfied with the facility.	6	1.00	Accepted
	Q10. I like to exercise at the facility.	6	1.00	Accepted
	Q11. I am satisfied with the exercise experience at the facility.	6	1.00	Accepted
	Q12. I feel comfortable when exercising at the facility.	6	1.00	Accepted
Reuse intention	Q13. I want to continue exercising at the facility.	6	1.00	Accepted
	Q14. When considering exercise, I think of the facility first.	6	1.00	Accepted
	Q15. I want to go back to the facility and exercise.	6	1.00	Accepted
	Q16. I want to actively participate in exercise at the facility.	6	1.00	Accepted
S-CVI/Ave ^b^ = 0.92				

I-CVI, item level content validity index; S-CVI/Ave, scale level content validity index average method. ^a^ Items with I-CVI ≥ 0.78 are considered excellent, according to Polit and Beck [26]. ^b^ S-CVI ≥ 0.90 is considered excellent, according to Polit and Beck [26].

**Table 3 ijerph-19-14606-t003:** Demographic characteristics of the respondents.

Variables		Frequency	Percentage (%)
Gender	Male	284	73.6
	Female	102	26.4
Marital status	Married	361	93.5
	Single	25	6.5
Age	65–74	361	93.5
	75–84	11	2.8
	≥85	14	3.6
Academic background	Graduated from high achool or less	140	36.3
	Graduated from college	189	48.9
	Graduated from graduate school or higher	57	14.8
Health status	Very healthy	21	5.4
	Healthy	365	94.6
	Average	0	0
	Unhealthy	0	0
	Very unhealthy	0	0
Use of facilities	Once a week or less	194	50.3
	2–3 times per week	132	34.2
	4–5 times per week	44	11.4
	6 or more times per week	16	4.1
Total		386	100

**Table 4 ijerph-19-14606-t004:** Differences in mechanistic service factors according to demographic characteristics.

Variables		Frequency	Mean	Standard Deviation	t/F	*p*	*Scheffe*
Gender	Male	284	3.57	0.638	1.656	0.100	
	Female	102	3.40	0.930			
Marital status	Married	361	3.56	0.719	3.809 ***	0.001	
	Single	25	3.01	0.690			
Health status	Healthy	365	3.53	0.642	0.138	0.892	
	Very healthy	21	3.48	1.648			
Age	65–74	361	3.53	0.696	0.307	0.736	
	75–84	11	3.36	0.722			
	≥85	14	3.48	1.369			
Academic background	Graduated from high school or less ^a^	140	3.25	0.807	22.260 ***	0.000	c > b > a
	Graduated from college ^b^	189	3.60	0.618			
	Graduated from graduate school or higher ^c^	57	3.94	0.611			
Use of facilities	Once a week or less ^a^	194	3.39	0.708	18.948 ***	0.000	d > a,b,c
	2–3 times per week ^b^	132	3.61	0.680			
	4–5 times per week ^c^	44	3.44	0.674			
	6 or more times per week ^d^	16	4.69	0.285			

*** *p* < 0.001.

**Table 5 ijerph-19-14606-t005:** Differences in humanistic service factors according to demographic characteristics.

Variables		Frequency	Mean	Standard Deviation	t/F	*p*	*Scheffe*
Gender	Male	284	2.73	0.898	4.275 ***	0.000	
	Female	102	2.35	0.706			
Marital status	Married	361	2.61	0.872	−1.358	0.175	
	Single	25	2.85	0.758			
Health status	Healthy	365	2.69	0.839	5.892 ***	0.000	
	Very healthy	21	1.59	0.657			
Age	65–74 ^a^	361	2.71	0.818	34.616 **	0.003	a > b > c
	75–84 ^b^	11	1.97	0.623			
	≥85 ^c^	14	1.14	0.361			
Academic background	Graduated from high school or less ^a^	140	2.44	0.856	5.799 ***	0.003	b > a
	Graduated from college ^b^	189	2.77	0.838			
	Graduated from graduate school or higher ^c^	57	2.62	0.912			
Use of facilities	Once a week or less ^a^	194	2.40	0.805	15.198 ***	0.000	b > d c > a
	2–3 times per week ^b^	132	2.91	0.873			
	4–5 times per week ^c^	44	2.97	0.551			
	6 or more times per week ^d^	16	2.04	1.167			

** *p* < 0.05, *** *p* < 0.001.

**Table 6 ijerph-19-14606-t006:** Differences in user satisfaction according to demographic characteristics.

Variables		Frequency	Mean	Standard Deviation	t/F	*p*	*Scheffe*
Gender	Male	284	3.37	0.614	2.049 *	0.042	
	Female	102	3.16	0.952			
Marital status	Married	361	3.31	0.738	−0.757	0.450	
	Single	25	3.42	0.461			
Health status	Healthy	365	3.36	0.664	6.033 ***	0.000	
	Very healthy	21	2.24	0.850			
Age	65–74 ^a^	361	3.38	0.635	66.078 **	0.000	a,b > c
	75–84 ^b^	11	3.41	0.340			
	≥85 ^c^	14	1.43	0.514			
Academic background	Graduated from high school or less ^a^	140	3.18	0.833	3.712 **	0.025	b,c > a
	Graduated from college ^b^	189	3.39	0.590			
	Graduated from graduate school or higher ^c^	57	3.39	0.798			
Use of facilities	Once a week or less ^a^	194	2.98	0.682	15.198 ***	0.000	b,c > a
	2–3 times per week ^b^	132	3.71	0.460			
	4–5 times per week ^c^	44	3.61	0.657			
	6 or more times per week ^d^	16	3.33	1.106			

* *p* < 0.01, ** *p* < 0.05, *** *p* < 0.001.

**Table 7 ijerph-19-14606-t007:** Differences in reuse intention according to demographic characteristics.

Variables		Frequency	Mean	Standard Deviation	t/F	*p*	*Scheffe*
Gender	Male	284	3.35	0.686	3.060 **	0.003	
	Female	102	3.06	0.886			
Marital status	Married	361	2.61	0.872	−1.396	0.163	
	Single	25	2.85	0.758			
Health status	Healthy	365	2.69	0.839	4.661 ***	0.000	
	Very healthy	21	1.59	0.657			
Age	65–74 ^a^	361	2.71	0.818	56.267 ***	0.000	a,b > c
	75–84 ^b^	11	1.97	0.623			
	≥85 ^c^	14	1.07	0.271			
Academic background	Graduated from high school or less ^a^	140	3.24	0.867	0.444	0.642	
	Graduated from college ^b^	189	3.31	0.555			
	Graduated from graduate school or higher ^c^	57	3.25	1.001			
Use of facilities	Once a week or less ^a^	194	2.99	0.771	23.024 ***	0.000	b,c > a
	2–3 times per week ^b^	132	3.59	0.507			
	4–5 times per week ^c^	44	3.59	0.717			
	6 or more times per week ^d^	16	3.25	1.000			

** *p* < 0.05, *** *p* < 0.001.

**Table 8 ijerph-19-14606-t008:** Pearson correlation analysis between variables.

Variables		1	2	3	4
Service factors	1. Mechanistic service factors	1			
	2. Humanistic service factors	0.254 **	1		
3. User satisfaction		0.488 **	0.625 **	1	
4. Reuse intention		0.365 **	0.507 **	0.868 **	1

** *p* < 0.05.

**Table 9 ijerph-19-14606-t009:** Effect of service factors on user satisfaction.

Variables	Unstandardized Coefficient (B)	Standard Error (SE)	Standardized Coefficient (β)	T	*p*
(Constant)	0.909	0.137		6.634	0.000
Mechanistic service factors	0.350	0.037	0.352	9.494 ***	0.000
Humanistic service factors	0.447	0.031	0.535	14.420 ***	0.000

F = 196.380, *** *p* < 0.001, R^2^ = 0.506.

**Table 10 ijerph-19-14606-t010:** Effect of service factors on reuse intention.

Variables	Unstandardized Coefficient (B)	Standard Error (SE)	Standardized Coefficient (β)	T	*p*
(Constant)	1.342	0.168		7.982	0.000
Mechanistic service factors	0.262	0.045	0.253	5.794 ***	0.000
Humanistic service factors	0.386	0.038	0.443	10.146 ***	0.000

F = 88.869. *** *p* < 0.001, R^2^ = 0.317.

**Table 11 ijerph-19-14606-t011:** Effect of user satisfaction on reuse intention.

Variables	Unstandardized Coefficient (B)	Standard Error (SE)	Standardized Coefficient (β)	T	*p*
(Constant)	0.277	0.090		3.090	0.002
User satisfaction	0.905	0.026	0.868	34.200 ***	0.000

F = 1169.670, ***, *p* < 0.001, R^2^ = 0.753.

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
