# Peer review of "Effects of Service Quality Characteristics of Neighborhood Sports Facilities on User Satisfaction and Reuse Intention of the Elderly during the COVID-19 Pandemic"

_ijerph, 2022, doi:10.3390/ijerph192114606_

Round 1
Reviewer 1 Report (Previous Reviewer 2)
Thank you for the opportunity to review this manuscript. I am satisfied with the revised statistical analysis and elaboration on manuscript sections. I advice to shorten Conclusion to one paragraph and include Limitations to the Discussion section.
Author Response
Thank you for your review.
In conclusion, this study confirmed that the mechanistic and humanistic service factors of neighborhood sports facilities during the COVID-19 pandemic had a positive effect on the elderly's user satisfaction and reuse intention. If mechanistic and humanistic service factors are not improved, the elderly may not be satisfied with neighborhood sports facilities or may not reuse them in the post-corona era. Therefore, this study has some implications that it is necessary to maintain the service quality factors of the neighborhood sports facilities to increase user satisfaction and reuse intention. However, this study has a limitation in that it did not analyze potential mediating or moderating variables in relation to service factors of the neighborhood sports facilities, user satisfaction, and reuse intention. Therefore, the need to identify potential mediating or moderating effects among the variables using structural equation modeling in follow-up studies is suggested.
Regards
Cho et al.

Reviewer 2 Report (Previous Reviewer 1)
N/A
Author Response
Thank you for your review.
In conclusion, this study confirmed that the mechanistic and humanistic service factors of neighborhood sports facilities during the COVID-19 pandemic had a positive effect on the elderly's user satisfaction and reuse intention. If mechanistic and humanistic service factors are not improved, the elderly may not be satisfied with neighborhood sports facilities or may not reuse them in the post-corona era. Therefore, this study has some implications that it is necessary to maintain the service quality factors of the neighborhood sports facilities to increase user satisfaction and reuse intention. However, this study has a limitation in that it did not analyze potential mediating or moderating variables in relation to service factors of the neighborhood sports facilities, user satisfaction, and reuse intention. Therefore, the need to identify potential mediating or moderating effects among the variables using structural equation modeling in follow-up studies is suggested.
Regards
Cho et al.

This manuscript is a resubmission of an earlier submission. The following is a list of the peer review reports and author responses from that submission.
Round 1
Reviewer 1 Report
This article explores the relationship between elders' satisfaction and reuse intention on neighborhood sports facilities. The paper is well written and easy to understand. My concerns on statistical analysis is listed below:
(1) In Tables 5 and 6, the authors applied regression analysis on the key variables such as reuse intention and satisfaction. However, the model does not adjust for potential confounders, such as gender. It would be much more convincing if the regression analysis include those variables, and it may lead to more discoveries, e.g., female's reuse intention is more sensitive to satisfaction compared to males.
(2) Table 2: The exploratory factor analysis helps people to decide how to cluster the item questions. It seems to me that the authors are doing confirmatory factor analysis - already given the structure of the questionnaire, e.g., the first 4 items are measuring Machinistic service factors, the authors check the loadings to see whether the structure is correct
(3) Row 125: '. It can be interpreted to mean that elderly users feel the improvement in information provision and medical support services'. I don't understand the interpretation of ANOVA results. The F=175.871 and pvalue<0.05 can only indicate that the scores differ significantly across the 4 sub-factors
Author Response
Thank you very much for your comments.
The content is reflected in the text as follows.
(1) Revised sentence
Meanwhile, the elderly living in small towns or rural areas where there are not many indoor sports facilities may have a high possibility of using neighborhood sports facilities. In a follow-up study, it will be helpful to generalize the research results by separately approaching the elderly in rural and urban areas. Also, examining the differences in the perceptions of the elderly about neighborhood living facilities according to gender could provide more meaningful research results.
(2) We did confirmatory factor analysis. Sorry for our mistake.
-->Confirmatory factor analysis was performed to analyze the reliability and validity of the research tool, and Cronbach’s α was also extracted for the analysis (Table 1)
(3)Post hoc results are presented in the text.
After ANOVA test for sub-factors of service quality, Bonferroni test was performed, which is the simplest post hoc analysis. When examining the elderly's perception of sub-factors of service quality of neighborhood sports facilities, the machinistic service factor had a mean score of 3.563±0.459, and the humanistic service factor had a mean score of 3.266±0.397. Besides, user satisfaction was 3.629±0.39, and reuse intention was 3.806±0.555. Moreover, there was a statistically significant difference (F=175.871, p<0.001) among safety (3.864±0.625), program (3.523±0.512), information provision (3.187± 0.531), and medical support (2.493±0.499), which are sub-factors of the independent humanistic service factor variable. According to Bonferroni test, the differences between the four sub-factors were still significant. This may imply that elderly users experienced an improvement in information provision and medical support services.

Reviewer 2 Report
Thank you for the opportunity to review this manuscript. The paper is interesting, however, I feel like several parts of the paper lack the appropriate level of clarity. See specific comments below:
1. Line 4 – “copies who responded insincerely” – how it was assessed?
2. Line 49 – “the effect of the service quality of neighborhood sport facilities on the satisfaction” – I suggest to state clear characteristics of the term ‘service quality’ in the introduction. It is not clear what exactly will be assessed in the study while reading introduction or abstract.
3. Line 56 – “H1: The service quality of neighborhood sports facilities shall affect user satisfaction.” – I suggest to be more specific and state the direction of the anticipated association (e.g. increase, decrease, etc.) How exactly it will affect user satisfaction. I have similar suggestions for H2 and H3.
4. Line 67 – “26 questionnaires that were answered insincerely were 67 excluded” – How it was assessed?
5. How individuals were selected for the study exactly? What was selection algorithm?
6. I suggest to include the questionnaire to the Appendix.
7. I suggest to elaborate more on statistical analysis section to ensure reproducibility of this study. I suggest to move details regarding factor analysis to Methods section from Results section.
8. Table 3. It was not clear to me right away what those numbers actually represent. Could you provide more details in Footnote of the Table 3.
9. Section 3.3. As my previous suggestion, I suggest to add the hypothesized direction of the association. It will be easier to read and follow. What type of dependent and independent variables were used (categorical, continuous). Did you control for any other factors in the regression model? If there were multiple independent variables/control factors in the regression how did you assess multicollinearity (correlation between those factors)?
10. Line 170 – “repurchase intention” - What does it mean in this manuscript? It is the first time when it was used in the manuscript text, and was not defined earlier or stated in H3 earlier. Earlier it was reuse. Please, provide clear definition of these terms regarding your study.
11. Line 181 – “This study divided service quality into machinistic and humanistic service factors, and it was confirmed that elderly users perceived machinistic service factors (3.563±0.459) higher than humanistic service factors (3.266±0.397).” How significant is this difference regarding real world implementation? 3.56 and 3.266 seem like relatively close numbers. Maybe use percentages to visualize the findings in more effective way.
12. Line 184 “In particular, elderly users perceived that the provision of facility information (3.187±0.531) and the provision of medical service (2.493±0.499) was insufficient.” – How it was measured? What is a threshold between sufficient and insufficient level? Insufficient compared to what? It is not clear to me.
13. I suggest to elaborate more on Strengths and limitations section and make it a part of the Discussion section. How other sample characteristics or methodologic approaches, besides metropolitan area of residence, might affect the study results?
14. I suggest to elaborate more on the idea how the findings of this study advance the existing knowledge in this field.
Author Response
Thank you for the wonderful comments. We are so sorry for our mistakes.
We did our best to fix all of them
- Line 4 – “copies who responded insincerely” – how it was assessed?
Among them, 26 responses that were deemed insincere (e.g., two or more answers missing) were excluded, resulting in 154 valid responses for analysis. - Line 49 – “the effect of the service quality of neighborhood sport facilities on the satisfaction” – I suggest to state clear characteristics of the term ‘service quality’ in the introduction. It is not clear what exactly will be assessed in the study while reading introduction or abstract.
Service quality related to neighborhood sports facilities refers to machinistic (objective) quality factors and humanistic (subjective) quality factors provided to users at the facility. This study aims to provide basic data for the expansion of facilities that contribute to the health of the elderly by examining the effect of the service quality of neighborhood sport facilities on the satisfaction and reuse intention of elderly users.
- Line 56 – “H1: The service quality of neighborhood sports facilities shall affect user satisfaction.” – I suggest to be more specific and state the direction of the anticipated association (e.g. increase, decrease, etc.) How exactly it will affect user satisfaction. I have similar suggestions for H2 and H3.
H1: The service quality of neighborhood sports facilities shall have a positive effect on user satisfaction.
H2: The service quality of neighborhood sports facilities shall have a positive effect on reuse intention.
H3: User satisfaction shall have a positive effect on reuse intention.
. 4. Line 67 – “26 questionnaires that were answered insincerely were excluded” – How it was assessed?
Among them, 26 responses that were deemed insincere (e.g., two or more answers missing) were excluded, resulting in 154 valid responses for analysis.
- How individuals were selected for the study exactly? What was selection algorithm?
In this study, random sampling was performed on the elderly living in Seoul.
(Selection algorithm--> deleted) - I suggest to include the questionnaire to the Appendix.
Yes, we did. Check the appendix please. - I suggest to elaborate more on statistical analysis section to ensure reproducibility of this study. I suggest to move details regarding factor analysis to Methods section from Results section.
Yes, we did. Check "2.4 Confirmatory factor analysis" please. - Table 3. It was not clear to me right away what those numbers actually represent. Could you provide more details in Footnote of the Table 3.
Note: The values of the mean and standard deviation are the results of the participants answering each item using a Likert 5-point scale. (S.D. : Standard Deviation)
- Section 3.3. As my previous suggestion, I suggest to add the hypothesized direction of the association. It will be easier to read and follow. What type of dependent and independent variables were used (categorical, continuous). Did you control for any other factors in the regression model? If there were multiple independent variables/control factors in the regression how did you assess multicollinearity (correlation between those factors)?
Multicollinearity is a problem that can occur with regression analysis when there is a high correlation of at least one independent variable with a combination of the other independent variables. The variance inflation factor (VIF) and tolerance are two closely related statistics for diagnosing collinearity in multiple regression. In this study, there is no significant multicollinearity (VIF<10). - Line 170 – “repurchase intention” - What does it mean in this manuscript? It is the first time when it was used in the manuscript text, and was not defined earlier or stated in H3 earlier. Earlier it was reuse. Please, provide clear definition of these terms regarding your study.
Sorry. The word repurchase intention has been deleted. We used reuse intention. - Line 181 – “This study divided service quality into machinistic and humanistic service factors, and it was confirmed that elderly users perceived machinistic service factors (3.563±0.459) higher than humanistic service factors (3.266±0.397).” How significant is this difference regarding real world implementation? 3.56 and 3.266 seem like relatively close numbers. Maybe use percentages to visualize the findings in more effective way.
This study divided service quality into machinistic and humanistic service factors, confirming that elderly users perceived machinistic service factors (71.3%, 3.563±0.459) to be higher than humanistic service factors (65.3%, 3.266±0.397). - Line 184 “In particular, elderly users perceived that the provision of facility information (3.187±0.531) and the provision of medical service (2.493±0.499) was insufficient.” – How it was measured? What is a threshold between sufficient and insufficient level? Insufficient compared to what? It is not clear to me.
These are sub-factors lower than the average score of humanistic factors of 3.266.
--> In particular, elderly users perceived the provision of facility information (3.187±0.531) and the provision of medical service (2.493±0.499) as insufficient. This suggest the crucial role of exercise professionals in the operation of neighborhood sports facilities - I suggest to elaborate more on Strengths and limitations section and make it a part of the Discussion section. How other sample characteristics or methodologic approaches, besides metropolitan area of residence, might affect the study results?
Fourth, this study has a limitation in that it was conducted on the elderly living in Seoul during the COVID-19 pandemic. The elderly living in Seoul have many opportunities to use fitness centers and indoor gymnasiums. Therefore, when the COVID-19 pandemic is over, the elderly in Seoul can exercise indoors rather than using the neighborhood sports facilities. Meanwhile, the elderly living in small towns or rural areas where there are not many indoor sports facilities may have a high possibility of using neighborhood sports facilities. In a follow-up study, it will be helpful to generalize the research results by separately approaching the elderly in rural and urban areas. Also, examining the differences in the perceptions of the elderly about neighborhood living facilities according to gender could provide more meaningful research results.
- I suggest to elaborate more on the idea how the findings of this study advance the existing knowledge in this field.
Fourth, this study has a limitation in that it was conducted on the elderly living in Seoul during the COVID-19 pandemic. The elderly living in Seoul have many opportunities to use fitness centers and indoor gymnasiums. Therefore, when the COVID-19 pandemic is over, the elderly in Seoul can exercise indoors rather than using the neighborhood sports facilities. Meanwhile, the elderly living in small towns or rural areas where there are not many indoor sports facilities may have a high possibility of using neighborhood sports facilities. In a follow-up study, it will be helpful to generalize the research results by separately approaching the elderly in rural and urban areas. Also, examining the differences in the perceptions of the elderly about neighborhood living facilities according to gender could provide more meaningful research results.

Round 2
Reviewer 1 Report
The authors do not address the first concern that no confounders are controlled in the regression analysis. It would be easy to address this by adding age, gender, etc. below the primary variables (Machinistic service factors, Humanistic factors). Since the p-values of primary variables are so small, I feel it would not impact the conclusion while significance improve the validity of this analysis. It would be great that the authors can add them. However, since the authors include some sentences as limitations, the current one may be OK.
Author Response
Dear Reviewr
1. We conducted the survey again and performed statistical processing again.
2. We address this by adding age, gender, etc.
3. We resubmit the thesis after major revision.
Thank you

Reviewer 2 Report
Thank you for the opportunity to review the revised version.
I have concerns with the Questionnaire content (Supplement). English translation of those questionnaire items assumes Yes/No answers and not really Likert scale.
In addition, items in Questionnaire (at the Supplement) does not match Table 3.
For example:
I want to go back to the facility and exercise. (Table 3)
21. Do you want to revisit the neighborhood sports facilities?
1) strongly disagree 2) disagree 3) Neutral 4) agree 5) strongly agree
Or
Is information on the facility provided? (Table 3).
13. Do you think you will be provided with information about neighborhood living and sports
facilities?
1) strongly disagree 2) disagree 3) Neutral 4) agree 5) strongly agree
Using English version, those questions and statements may represent different data which may be collected.
I am not clear why those differences exist between Table 3 and Supplement, and what kind of data was actually collected with original survey tool.
Author Response
Dear Reviewer
1. We judged that the survey and statistical processing were insufficient, so we conducted the survey again and processed the statistics again.
2. We can send you the Excel file that has been investigated and statistically processed.
3. There were errors in the process of translating the survey conducted in Korean into English, and all of them were corrected.
------------------------------------------------------------------------------
I have concerns with the Questionnaire content (Supplement). English translation of those questionnaire items assumes Yes/No answers and not really Likert scale.
==>Major revision was done
In addition, items in Questionnaire (at the Supplement) does not match Table 3.
==>Matched
